# Reported Prevalence and Nutritional Management of Functional Constipation among Young Children from Healthcare Professionals in Eight Countries across Asia, Europe and Latin America

**DOI:** 10.3390/nu14194067

**Published:** 2022-09-30

**Authors:** Louise Naz West, Irina Zakharova, Koen Huysentruyt, Sze-Yee Chong, Marion M. Aw, Andy Darma, Badriul Hegar, Ruey Terng Ng, Mohammed Hasosah, Erick Toro-Monjaraz, Merih Cetinkaya, Chung-Mo Chow, Leilani Muhardi, Urszula Kudla, Dianne J. M. Delsing, Yvan Vandenplas

**Affiliations:** 1FrieslandCampina, 3818 LE Amersfoort, The Netherlands; 2Department of Pediatrics, Russian Medical Academy Continuous Professional Education of the Ministry of Health of Russian Federation, Moscow 125993, Russia; 3UZ Brussel, KidZ Health Castle, Vrije Universiteit Brussel (VUB), 1090 Brussels, Belgium; 4Department of Pediatrics, Hospital Raja Permaisuri Bainun, Ipoh 30450, Malaysia; 5Department of Paediatrics, Yong Loo Lin School of Medicine, National University of Singapore, Singapore 119077, Singapore; 6Department of Paediatrics, Khoo Teck Puat-National University Children’s Medical Institute, National University Health System, Singapore 119228, Singapore; 7Department of Pediatrics, Faculty of Medicine, Universitas Airlangga, Surabaya 60131, Indonesia; 8Department of Pediatrics, Faculty of Medicine, Universitas Indonesia, Jakarta 10430, Indonesia; 9Department of Pediatrics, University of Malaya, Kuala Lumpur 50603, Malaysia; 10Department of Pediatric, King Saud Bin Abdulaziz University for Health Sciences, Jeddah 14611, Saudi Arabia; 11King Abdullah International Medical Research Center (KAIMRC), Jeddah 11481, Saudi Arabia; 12Unit of Physiology and Gastrointestinal Motility, Gastroenterology and Nutrition Department, National Institute of Pediatrics, Mexico 04530, Mexico; 13Department of Neonatology, Health Sciences University, Basaksehir Cam and Sakura City Hospital, Istanbul 34480, Turkey; 14Virtus Medical Group, Hong Kong; 15FrieslandCampina AMEA, Singapore 039190, Singapore

**Keywords:** prevalence, nutritional management, functional constipation, toddler, survey, healthcare professionals

## Abstract

Background: The prevalence of functional constipation (FC) among children varies widely. A survey among healthcare professionals (HCPs) was conducted to better understand the HCP-reported prevalence and (nutritional) management of FC in children 12–36 months old. Methods: An anonymous e-survey using SurveyMonkey was disseminated via emails or WhatsApp among HCPs in eight countries/regions. Results: Data from 2199 respondents were analyzed. The majority of the respondents (65.9%) were from Russia, followed by other countries (Indonesia (11.0%), Malaysia (6.0%)), Mexico, KSA (5.1% (5.7%), Turkey (3.0%), Hong Kong (2.2%), Singapore (1.1%)). In total, 80% of the respondents (n = 1759) were pediatricians. The prevalence of FC in toddlers was reported at less than 5% by 43% of the respondents. Overall, 40% of the respondents reported using ROME IV criteria in > 70% of the cases to diagnose FC, while 11% never uses Rome IV. History of painful defecation and defecations < 2 x/week are the two most important criteria for diagnosing FC. In total, 33% of the respondents reported changing the standard formula to a specific nutritional solution, accompanied by parental reassurance. Conclusion: The most reported prevalence of FC in toddlers in this survey was less than five percent. ROME IV criteria are frequently used for establishing the diagnosis. Nutritional management is preferred over pharmacological treatment in managing FC.

## 1. Introduction

In 2016, the ROME Foundation introduced ROME IV criteria to replace the previous ROME III criteria and updated the definition of Functional Constipation (FC) in children. In order to diagnose FC, at least two of the following criteria should be present for non-toilet-trained children: (i) two or fewer defecations per week, (ii) history of excessive stool retention, (iii) history of painful or hard bowel movements, (iv) history of large diameter stools, (v) presence of a large fecal mass in the rectum [1]. In toilet-trained children, the following criteria are added: (i) at least one episode per week of incontinence after the acquisition of toileting skills, (ii) a history of large diameter stools which may obstruct the toilet [1].

Based on epidemiological studies, the prevalence of FC in young children has been reported to vary widely worldwide between 1.3% and 26.8% [2]. However, only a limited number of studies have reported specifically on the prevalence in toddlers (1–3 years of age) [3,4,5,6]. In addition, there are also very limited data regarding the prevalence and management of FC in toddlers in Asia and Latin America.

Despite the fact that the ROME IV criteria have been implemented for more than 6 years, the level at which these criteria are known to and applied by healthcare professionals (HCPs) in various countries remains largely unknown. In addition, the way in which HCPs manage FC in toddlers in daily practice is not widely reported. With this international survey among HCPs, we aimed to capture insights on (i) the estimated prevalence of FC in toddlers, (ii) the criteria applied for diagnosing FC and (iii) the preferred management.

## 2. Material and Methods

HCPs from eight countries/regions (Hong Kong, Indonesia, Kingdom of Saudi Arabia, Malaysia, Mexico, Russia, Singapore and Turkey) participated in an anonymous online survey using SurveyMonkey from August 2021 to 1 March 2022. The locations were selected based on the network of pediatric gastroenterologists of the initiators of the study (authors LM and YV).

Information on the demographics, diagnostic criteria, prevalence and management of FC in toddlers was asked in the survey. Before disseminating it electronically, the questionnaire was piloted in Malaysia, Indonesia and Mexico to determine the quality, understandability, relevance and suitability of the questions and to accommodate the local/regional variation in clinical practice. On average, 90% of the survey questions were consistently implemented across the countries/regions. The questionnaire consists of 30–46 questions depending on the participating countries (Appendix A). For this manuscript, only the questions regarding toddlers were presented. An example of the full questionnaire that was distributed in Russia, which provided the greatest number of respondents, can be seen in Appendix B.

Approval from institutional ethics review committees from each participating center was obtained prior to the survey’s implementation. The name of the study’s sponsor (FrieslandCampina) was mentioned in the introduction of the electronic questionnaire.

The target respondents were pediatricians, pediatric gastroenterologists and general practitioners (GP) who were contacted based on the network of the principal investigators in each country. The link to the online survey was sent randomly to HCPs engaged in public or private practices as potential participants via WhatsApp messages or emails based on the available network of the investigators. Each country was targeted to reach out to 100 respondents except for countries with a limited number of HCPs such as Singapore, Hong Kong and Turkey. Reminders were sent when the target number of respondents was not achieved.

IBM SPSS System for Windows, version 24 (IBM Corp., Armonk, NY, USA) was used to implement statistical analyses. Responses were excluded if the respondent did not see any constipation cases in their practice in the last week and/or if they completed less than 50% of the questionnaire. Descriptive statistics were computed to present qualitative data as the absolute number and percentages (the number for each response as a nominator and the total number of responses as the denominator) to depict the actual number of responses. The denominator can be different for each question depending on the number of responses and the missing values. The percentages were the round-up to one decimal point. Chi-Square or Fisher’s exact test was used to examine differences between categories (type of professions, years of practice and regions), with *p*-values < 0.05 considered statistically significant.

## 3. Results

### 3.1. General Information of the Respondents

A total of 2596 healthcare professionals responded to the survey, with 2199 (85%) of the respondents included in the data analysis (Figure 1). The dropout was due to the exclusion criterion of having incomplete responses. The majority of respondents were from: Russia (65.9%, 1449), followed by other countries (Indonesia (11.0%, 242), Malaysia (6.0%, 132), Mexico (5.7%, 125), Kingdom of Saudi Arabia (KSA) (5.1%, 113), Turkey (3.0%, 66), Hong Kong (2.2%, 49) and Singapore (1.1%, 23)), which were divided into three groups based on geographical areas whenever possible, namely region 1 (Asia consists of Indonesia, Malaysia, Hong Kong and Singapore), region 2 (Europe consists of Russia and Turkey), region 3 (rest of the world consists of KSA and Mexico (see Figure 1)).

Pediatricians represented the highest percentage of respondents (80.3%, 1759/2191), followed by pediatric gastroenterologists (7%, 154/2191), GP (6.3%, 139/2191) and other HCPs, including residents (6.3%, 139/2191 (Figure 2a)). Almost half (43.0%, 946/2195) of the respondents had more than 15 years of experience, while 24.0% (528/2195) had less than 5 years of experience and the rest were in between these periods (Figure 2b). In particular, around half of the respondents in Turkey (51.5%, 34/66) and Indonesia (49.6%, 120/242) had less than 5 years of experience, whereas, in the other countries, this percentage was much lower.

### 3.2. Diagnosis

#### 3.2.1. Reported Prevalence

Overall, almost half of the respondents (43.3%, 952/2189) reported estimating the prevalence of toddler FC as less than 5%.

This information was based on the estimated percentage of cases that the respondents saw during the last week before the survey was administered. There was a significant difference in reported prevalence across the three regions. More HCPs in regions 1 and 2 reported a prevalence of less than 5%, while a higher percentage of HCPs in region 3 (Mexico and KSA) reported a prevalence higher than 15% (Figure 3). There was no difference in reported prevalence among professions. Around 57.8% of respondents ((1273/2199) reported the peak age for FC in toddlers between the ages of 24 and 36 months.

#### 3.2.2. The Frequency of Use of ROME IV Criteria

The frequency of use of the ROME IV criteria for diagnosis of FC was determined in this survey as follows: ‘never’ (<10%), ‘rarely’ (10–30%), ‘sometimes’ (30–70%) and ‘almost always’ (>70%). Overall, 40.2% of the respondents (881/2199) reported that they almost always used the ROME IV criteria to establish the diagnosis of FC in toddlers, while around 10.7% (234/2199) reported never using the criteria.

More HCPs in region 2 (Turkey and Russia) and region 3 (Saudi Arabia and Mexico) reported almost always using the ROME IV criteria compared to the response from region 1 (Hong Kong, Indonesia, Malaysia and Singapore) (44.8% and 38.4% vs. 25.4%, respectively) (Table 1). There was a significant difference in the frequency of using ROME IV criteria among professions. Pediatricians reported almost always using the ROME IV criteria, while GP reported to rarely use them (*p* < 0.001).

Of all the respondents, 47.6% (1014/2131) reported that all the ROME IV criteria were important for the diagnosis of FC. Those who always used the ROME IV criteria were more likely to select all the ROME IV criteria equally important than those who never used them (*p* < 0.001). In addition, painful or hard bowel movements and two or fewer defecations per week were the two most often used criteria for diagnosis of FC (34.8% (741/2131) and 34.2% (729/2131), respectively).

#### 3.2.3. Warning Signs

Respondents were asked which warning signs of constipation they considered most important (abdominal distention, anal/sacral abnormalities, bloody/mucoid stools, failure to thrive, neurodevelopmental delay, vomiting). The format of this question was different in Russia, where the HCPs could only select one option, while in the other countries multiple responses could be given. Half of the respondents from Russia (51.2%, 740/1444), selected the “all of the above” option. In contrast, in the other countries, only 27.7% (383/1385) of all answers given were “all of the above.” HCPs in Russia considered the “presence of blood and mucus in the stool” (33.9%, 489/1444) as the most important warning sign, followed by “abdominal distention” (5.3%, 77/1444) and “failure to thrive” (3.3%, 48/1444). In the other countries, the most important warning sign was answered to be “abdominal distension” (15.9%, 220/1385), followed by “presence of blood and mucus in the stool” (14.2%, 196/1385) and “failure to thrive” (13.7%, 190/1385). No significant differences in the choice of most important warning signs were reported regardless of the way the questions were asked.

### 3.3. Nutrition Management of FC

The intervention with the highest percentage was changing the standard formula into a specific nutritional solution (40.2%, 835/2077), followed by parental reassurance (31.7%, 659/2077) and a pharmacological approach using lactulose (17.0%, 353/2077) (Figure 4). HCPs with more than 10 years of experience reported preferring to switch to a particular formula; on the other hand, HCPs with less than 10 years of experience reported preferring to perform parental reassurance as the first line of management for FC (*p* < 0.001).

#### 3.3.1. Specific Nutritional Solutions

The question on nutrition solutions for FC had the option of multiple answers in some countries and only a single answer in the rest (Table 2). In both cases, the first choice of a nutritional solution for toddlers was the same: standard milk that contains fiber (29.0%, 540/1862, and 19.6%, 79/403, multiple vs. single answer-countries, respectively). In the countries with multiple answers, the second choice was young child formula (YCF) containing synbiotics (15.6%, 63/403), followed by YCF containing probiotics or not using a nutritional solution in third place (both 15.4%, 62/403). On the other hand, in the group giving only one answer, the second and third most preferred options for the nutritional management of FC were “other solutions,” such as increasing fiber, water intake or other YCF solutions such as goat-based or magnesium-containing formula (12.5%, 232/1862), followed by no specific nutritional solution (11.8%, 219/1862), respectively. Around 5–8% of the respondents also reported a preference for using hydrolyzed formula (either extensive or partial).

#### 3.3.2. Functional Ingredients to Help Soften the Stool Consistency

HCPs were asked whether they had heard of fiber, prebiotics, probiotics, synbiotics, milk fat or hydrolyzed protein to help soften stool consistency. The format of the questions was different in the different country surveys. In Russia and Mexico, the respondents were asked to select a single functional ingredient which they considered most effective for softening stools. Fiber was rated the most important ingredient by a considerably higher percentage (60.2%, 944/1569) as compared to prebiotics (13.0%, 204/1569) and hydrolyzed protein (8.8%, 138/1569), which were rated as next most important. In addition, HCPs in Singapore, Indonesia, Hong Kong and Turkey were asked if they knew the ingredients to impact stool consistency by answering “yes”, “no” or “I don’t know.” The most reported known ingredients to impact stool consistency were probiotics (93.7%. 356/380). In contrast, less than half of the respondents indicated that milk fat (39.7%, 151/380) was able to impact stool consistency. The participants in Saudi Arabia were asked to rank the ingredients that could soften the stool consistency. They reported that fiber was their first choice (55.0%, 55/100), followed by prebiotics (14.0%, 14/100) and milk fat (12.0%, 12/100).

#### 3.3.3. The Most Effective Fiber Reported in Managing FC

Around 20–40% of the respondents reported that they did not know which fiber was the most effective in managing FC (Figure 5). Carob bean gum (32.5%, 665/2044), followed by inulin (18.5%, 378/2044) and FOS (14.9%, 305/2044), were the most reported effective ones in managing FC. There was a significant difference in the reported choice of most effective fibers by region. For example, the percentage of HCPs choosing galacto-oligosaccharides (GOS) was significantly higher in KSA and Mexico as compared to regions 2 and 1 (14.1%, 32/227 vs. 10.3%, 153/1482 and 4.5%, 20/444, *p* < 0.001, respectively).

## 4. Discussion

Previously reported prevalence of FC among children aged 13–48 months based on primary data collected widely ranges from 1.3% to 26% [2]. In the current survey, most HCPs reported a less wide range to a maximum of 15%. This was consistent with the country’s specific prevalence data reported earlier in earlier epidemiological studies (Table 3).

FC prevalence in Hong Kong, Indonesia, Mexico, Russia, Singapore and Turkey among toddlers had not been previously reported. In these cases, HCPs reports of toddler prevalence of FC from our study could be utilized as a proxy for primary FC prevalence. This is because we used the “wisdom of crowds” principle to estimate the prevalence of FC in toddlers [9]. This concept is based on the idea that large groups of people collectively have a more correct estimation than individual experts as long as the quality of the crowd is ensured and individuals are able to answer independently from each other. As we limited the survey to HCPs that treated at least one child with FC in the past week and the ROME IV criteria were almost always used by most of the respondents, we can be confident about the quality of the opinion of the survey participants in their estimation of the prevalence.

The peak reported incidence of FC in toddlers in the HCP practice was reported to be 24–36 months of age. Earlier studies reported that toilet training [10] and fussy eating [11] could be linked to an increased occurrence of FC in this age group [12].

In 2016, the ROME IV criteria were introduced, which make a distinction between toilet-trained and -untrained toddlers under 2.5 years old compared to the previous ROME III criteria [1]. Nevertheless, the extent to which HCPs are aware of the criteria and use them is largely unknown. In this survey, around 40% of respondents reported almost always using the ROME IV criteria for the diagnosis of constipation. Although the number is higher than the previously reported information among pediatricians and pediatric gastroenterologists in South Korea [13], there is also a need to constantly educate healthcare professionals, particularly GP, on ROME IV criteria as diagnostic tools for FC. The reported frequency of use of the ROME IV criteria also differed by region, with Asian countries reporting less use of the criteria. In addition, HCPs with less than 5 years of experience tended to follow the guidelines in managing FC as compared to those with more years of experience. Thus, active continuing medical education in Asia and in more experienced HCPs are of importance.

Almost half of the HCPs in this survey reported using all the ROME IV criteria and considered them equally important. Painful or hard bowel movement and <x2/week defecation were the next top two criteria for constipation diagnosis. That HCPs seem to value these two criteria the most is corroborated by findings from another study in which 87% of pediatricians suspected constipation in children over 6 months of age when there was a decrease in the frequency of bowel movement, and 83% when hard stools were reported [13,14].

The evaluation of warning signs for constipation in children, such as abdominal distention, anal scars and failure to thrive, is critical in the diagnosis of constipation [15] because it can help point to an organic cause of constipation [16]. In this survey, regardless of the format of the question, the majority of HCPs did not have a preference of a single warning sign but regarded all (abdominal distention, anal/sacral abnormalities, bloody/mucoid stools, failure to thrive, neurodevelopmental delay, vomiting) as equally important.

In general, the initial management of constipation is recommended to include an adequate intake of fibers and fluids, as well as physical activity followed by pharmacological treatments if needed [17]. Furthermore, HCPs can optimize the management of constipation by providing extensive and recent parental education and reassurance [18]. In our survey, the HCPs, especially those with more than 15 years of experience, preferred nutritional management, such as changing the standard formula to a specific nutrition solution, followed by the reassurance of the parents over a pharmacological treatment. The preference for specific nutrition solutions varied from those containing fibers to the use of (extensive and partial) hydrolyzed protein formulas in accordance with the availability of the solution in each participating center. The practice to change to specific nutrition solution is in line with the preference of HCP previously reported in Indonesia [14] but in contrast with the recent results of the HCPs survey in Korea which reported the preference to use a laxative [13].

Fiber was reported to be the most important ingredient in softening stool according to the HCPs in the majority of countries in this survey, which is in line with the general reported knowledge [19]. However, when asked which fiber was believed to be the most beneficial in managing constipation, around 20–40% of the respondents in all regions answered they did not know, with a higher percentage in Asian countries. The most effective fiber reported in this survey was carob bean gum (CBG). This is an interesting finding given that CBG is commonly used as a thickening agent in infant formulas to prevent regurgitation and improve reflux symptoms and not for managing constipation [20]. Therefore, in practice, CBG may also be used for the treatment of other functional gastrointestinal disorders, such as constipation.

The respondents also reported inulin and FOS as the next best fibers to manage constipation. FOS and inulin have been shown in numerous studies to soften stool and/or increase stool consistency in infants and young children [21,22,23,24,25,26,27]. Unlike in the other regions, GOS were chosen as the second-best fiber to reduce constipation symptoms in Saudi Arabia and Mexico. This may reflect the wide application and therefore knowledge of GOS in YCF in those countries. Indeed, GOS supplementation has been reported to have functional effects and benefits for specific gastrointestinal improvements in children [28,29,30]. Our findings suggest that HCPs are aware of the potential benefits of different fibers in relieving child constipation, but there is room for more education on the type of fibers.

To the best of our knowledge, this study is the first multi-country survey conducted to reflect the diagnosis and management of constipation of toddlers by HCPs of different expertise levels across the regions. Nonetheless, our research has a few limitations. The number of respondents from each country was unequal among the participating countries. This could be attributed to the distribution of the survey, which was limited to those within a specific network, as well as people who are experienced with online surveys, and hence may be prone to selection bias. The data were dependent on respondents’ opinions, which could limit the survey’s objectiveness and applicability. Certain questions were phrased slightly differently in country-specific surveys, making exact comparisons slightly more challenging. There is also no information on stool-withholding behavior even though about 25% of constipated toddlers and young children typically exhibit this behavioral trait [31,32]. This behavior could further exacerbate the relapse of FC in later life [32].

On the other hand, the insights gained from this survey describe more reliable information about the knowledge of management in constipation than most studies based on reports by parents/caregivers. The sample size was quite large, with sufficient numbers of HCPs from each country participating in the study. Therefore, this survey adds valuable and reliable complementary information on early life constipation at a time when primary data collection was difficult due to the COVID-19 pandemic for each participating country or in totality.

## 5. Conclusions

The reported estimated prevalence of FC in toddlers seen by HCPs in eight countries was less than 5%. ROME IV criteria were frequently employed to establish the diagnosis of constipation. Specific nutritional solutions with functional fibers, pre- and probiotics and parental reassurance were reported to be preferred over pharmacological solutions in managing constipation. HCPs in Asia tended to use ROME IV criteria less frequently than those in other regions and had less knowledge of the different types of fibers to support the effective management of FC. The insights gained from this survey could further support the development of continuous medical education and tailored clinical care in the future.

## Figures and Tables

**Figure 1 nutrients-14-04067-f001:**
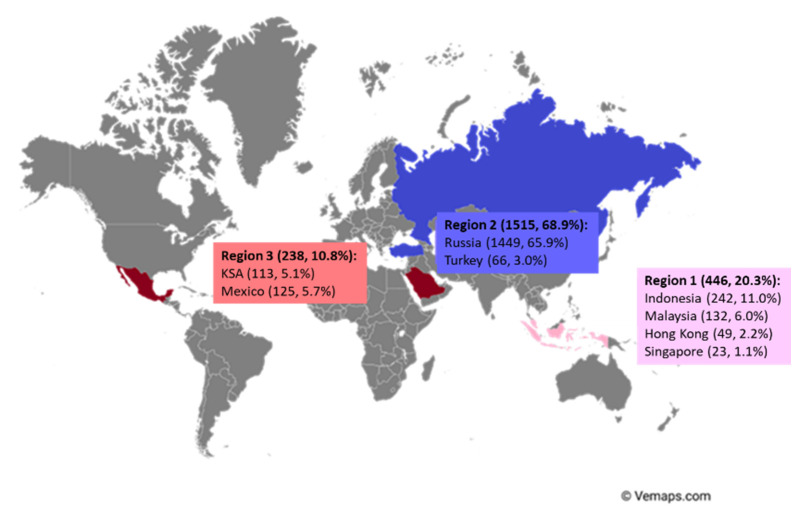
The distribution of survey respondents (number of respondents, % of total respondents).

**Figure 2 nutrients-14-04067-f002:**
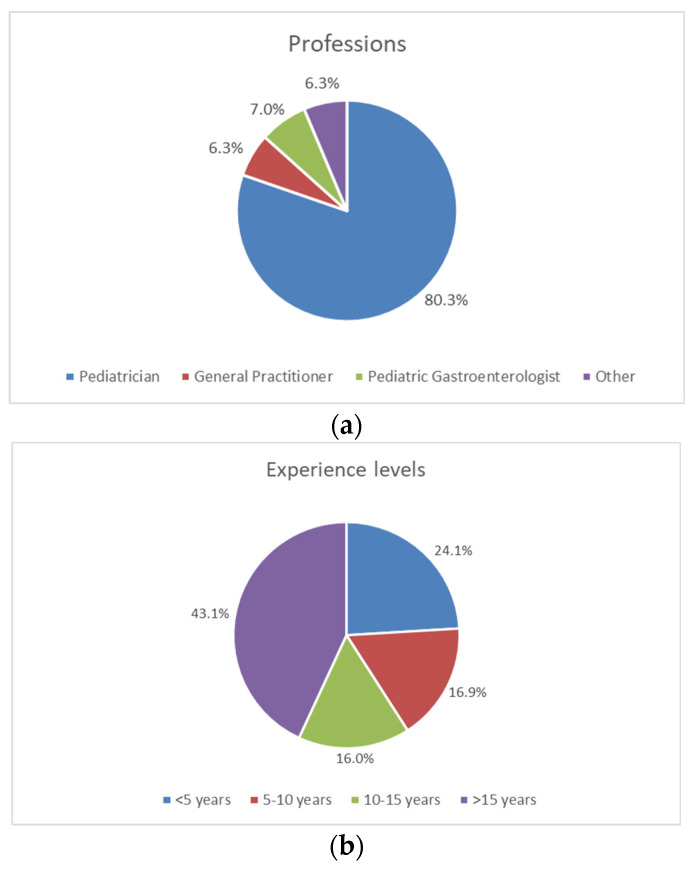
Demographics of the respondents. (**a**) Percentage by profession. (**b**) Percentage by years of experience.

**Figure 3 nutrients-14-04067-f003:**
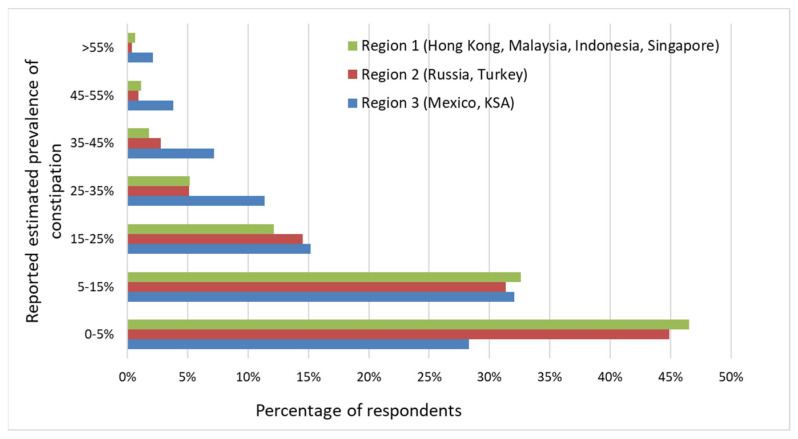
The estimated prevalence of toddler FC by region, reported by the respondents. Data are presented as the percentage of respondents estimating a certain prevalence percentage. Overall, the estimated prevalence was different between regions (*p* < 0.001).

**Figure 4 nutrients-14-04067-f004:**
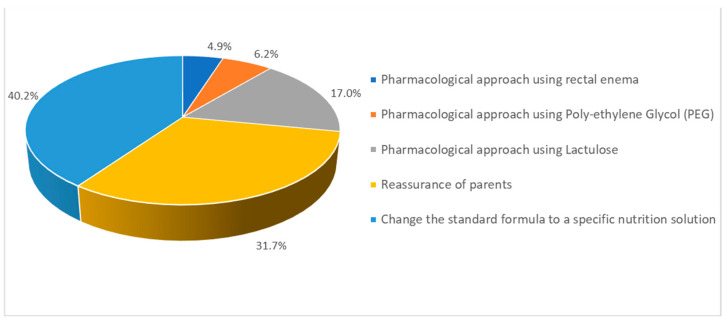
The reported first choice of management of FC. (% of total number of respondents).

**Figure 5 nutrients-14-04067-f005:**
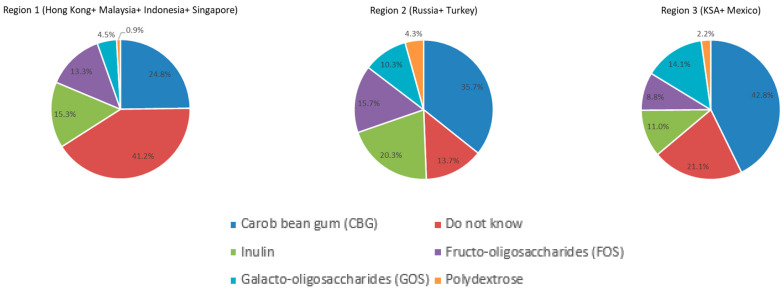
The most effective fiber reported for the management of FC by HCPs by region. (% of total number of respondents.).

**Table 1 nutrients-14-04067-t001:** The use of ROME IV criteria in the diagnosis of FC by regions.

	Region 1: Hong Kong + Malaysia + Indonesia + Singapore	Region 2: Russia + Turkey	Rest of World: KSA + Mexico	All
	N (%)	N (%)	N (%)	N (%)
Frequency of using Rome IV criteria in diagnosing FC in toddlers ^#,^*^,‡^				
• Almost always (>70%)	113/445 (25.4)	**677/1512 (44.8)**	**91/237 (38.4)**	**881/2194 (40.2)**
• Sometimes (30–70%)	**136/445 (30.6)**	418/1512 (27.6)	78/237 (32.9)	632/2194 (28.8)
• Rarely (10–30%)	116/445 (26.1)	288/1512 (19.0)	43/237 (18.1)	447/2194 (20.4)
• Never (<10%)	80/445 (18.0)	129/1512 (8.5)	25/237 (10.5)	234/2194 (10.7)
Rome IV criteria used most frequently (multiple responses ^a^				
• History of painful or hard bowel movement	**238/431 (55.2)**	408/1508 (27.1)	**95/192 (49.5)**	741/2131 (34.8)
• Two or fewer defecations per week	237/431 (55.0)	423/1508 (28.1)	69/192 (35.9)	729/2131 (34.2)
• History of excessive stool retention	159/431 (36.9)	365/1508 (24.2)	27/192 (14.1)	551/2131 (25.9)
• History of large diameter stools	99/431 (23.0)	189/1508 (12.5)	37/192 (19.3)	325/2131 (15.3)
• Presence of large fecal mass in the rectum	80/431 (18.6)	108/1508 (7.2)	39/192 (20.3)	227/2131 (10.7)
• All of the above	107/431 (24.8)	**840/1508 (55.7)**	67/192 (34.9)	**1014/2131 (47.6)**
• None of the above	5/431 (1.2)	34/1508 (2.3)	2/192 (1.0)	41/2131 (1.9)

N, number of respondents, which could be different for each question, Numbers in bold represent the highest response per region. ^#^ *p* < 0.001 between regions; * *p* < 0.001 between professions and their frequency of using ROME IV criteria; ^‡^
*p* < 0.001 between experience levels and their frequency of using ROME IV criteria; ^a^ multiple responses (respondents could choose multiple criteria).

**Table 2 nutrients-14-04067-t002:** The specific nutrition solutions the HCPs often use in managing FC in non-breastfed 1–3-year-old toddlers.

Nutritional Solutions in Managing FC	Hong Kong, Indonesia, Singapore, Russia, Mexico (Single Answer) ^#,^*	Malaysia, Turkey, KSA ^a^ (Multiple Answer)
Formula containing fiber	**540/1862 (29.0)**	**79/403 (19.6)**
Formula containing synbiotics	204/1862 (11.0)	63/403 (15.6)
No specific nutritional solution	219/1862 (11.8)	62/403 (15.4)
Formula containing probiotics	135/1862 (7.3)	62/403 (15.4)
Others	232/1862 (12.5)	23/403 (5.7)
Formula containing prebiotics	202/1862 (10.8)	46/403 (11.4)
Extensively hydrolyzed formula	149/1862 (8.0)	21/403 (5.2)
Partially hydrolyzed formula	135/1862 (7.3)	24/403 (6)
Soya-based infant formula	15/1862 (0.8)	3/403 (0.7)
Standard infant formula	31/1862 (1.7)	20/403 (5)

N, number of respondents, which could be different for each question; Numbers in bold represent the highest response per region. ^#^
*p* < 0.001 between professions and their choice of nutritional solutions for 1–3-year-old toddlers, * *p* < 0.001 between experience levels of HCPs and their choice of nutritional solutions for 1–3-year = old toddlers; ^a^ multiple response questions.

**Table 3 nutrients-14-04067-t003:** Overview of reported prevalence of FC based on epidemiological studies as compared to the current survey results.

	Epidemiological Studies				HCP Reported Estimation *
Country/Region	Diagnostic Criteria Used	Year	Age of Population	FC(%)	FC(%)
China [3]	ROME IV	2021	0–4 years	7	NA
Vietnam [7]	ROME IV	2022	0–48 months	4.6	NA
Malaysia [5]	ROME IV	2021	0–12 months	1.1%	<15%
Saudi Arabia [8]	ROME IV	2017	0–5 years	4.7%	5–15%
Singapore	NA				5–15%
Indonesia					<5%
Russia					<5%
Hong Kong					<15%
Mexico					<15%

* based on the highest reported prevalence.

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
