# Peer review of "Reported Prevalence and Nutritional Management of Functional Constipation among Young Children from Healthcare Professionals in Eight Countries across Asia, Europe and Latin America"

_nutrients, 2022, doi:10.3390/nu14194067_

Round 1
Reviewer 1 Report
This work is written very chaotically. It requires analysis and corrections because it causes a lot of errors and inaccuracies. The errors found in this article are listed below.
1. This review covers too few countries. Is there no problem in other countries, to which the authors of this article are devoted?;
Line 6 is "Marion M Aw 5,6 " should be "Marion M Aw5,6" (no spaces);
Lines 6-7 is "Badriul Hegar 8 " should be "Badriul Hegar8" (no spaces);
Line 10 and 22 too large spaces;
Line 10 is: "the Ministry" should be: "the Ministry"
Line 22 is “Gastrointestinal Motility. Gastroenterology ”should be“ Gastrointestinal Motility. Gastroenterology ";
Line 50 is "at least two of the following" should be "at least two of the following";
Line 51 must be deleted “;
Line 53 is "mass in the rectum 1 ." should be "mass in the rectum1.";
Line 55-56 is “obstruct the toilet 1 . "It should be" obstruct the toilet1. ";
Line 58 is "1.3% and 26.8% 2 ." it should be "1.3% and 26.8% 2.";
Line 59-60 is "(1-3 years of 59 age) 3-6 ." it should be “(1-3 years of 59 age)3-6.”;
Line 84 is' implementation. The name of the 'should be' implementation. The name of the ";
Line 90-93 is “Each country was targeted to reach out to 100 respondents except for countries with limited number of HCPs such as Singapore and Hong Kong. Reminders were sent when the target number of respondents was not achieved." And Turkey also has less than 100 respondents (3% - 66!);
Line 107-111 is “The drop-out was due to exclusion criteria of having incomplete responses Respondents were from the following countries / regions: Russia (65.9%, 1449), Indonesia (11.0%, 242), Malaysia (6.0%, 132 ), Mexico (5.7%, 125), Kingdom of Saudi Arabia (KSA) (5.1%, 113), Turkey (66, 3.0%), Hong Kong (2.2%, 49), Singapore (1.0%, 23)” and in line 33-34: (Russia (67%!), Indonesia (11%), Malaysia (6%), KSA (5%!), Mexico (5.7%), Turkey (3%), Hong Kong (2.2%), Singapore (1%)). - It needs to be standardized !!!;
Line 117-111 is “The drop-out was due to exclusion criteria of having incomplete responses Respondents were from the following countries / regions: Russia (65.9%, 1449), Indonesia (11.0%, 242), Malaysia (6.0%, 132 ), Mexico (5.7%, 125), Kingdom of Saudi Arabia (KSA) (5.1%, 113), Turkey (66, 3.0%), Hong Kong (2.2%, 49), Singapore (1.0%, 23)". Please count the% again for KSA, Hong Kong and Singapore!
2. I do not agree with the numbers and percentages in the whole work !!! You must check it all again! This misleads the person reading it.
3. Figure 1 is hardly legible!
Line 117 is “Pediatricians represented the highest percentage of respondents (80.3%, 1759/2199)” and should be: “Pediatricians represented the highest percentage of respondents (80.3%, 1766/2199!)”;
4. In Figure 2, “Profession”, the percentages need to be corrected. For example, the text says 80.3% and the chart 80%; 6.3% and the figure 6.32% !;
Line 128-129 is “Overall, the most often reported prevalence of toddler's FC was less than <5% (43.3%, 952/2189)” should be “Overall, the most often reported prevalence of toddler's FC was less than <5% (43.3%, 952/2199) ”;
Line 133 is "(Figure 2)" should be "(Figure 3)";
Line 137 is "(Figure 2)" should be "(Figure 3)";
Line 135 is "(1273/2199)" should be "((1275/2199)";
Line 141 and 143 you have to count the percentages !;
Line 146 is "Malaysia, and Singapore" should be "Malaysia and Singapore";
Line 146 is "44.8 % and 38.4% vs 25.4%" should be "44.8% and 38.4% vs 25.4%";
5. Figure 3 is hardly legible. I don't understand this drawing. Please describe it more in the text;
Line 150-155 is “Forty-eight percent of the respondents (1014/3628) reported that all of the ROME IV criteria were important for the diagnosis of FC. Those who always used the ROME IV criteria were more likely to select all of the ROME IV criteria equally important than those who never used them (p<0.001). In addition, painful or hard bowel movements and two or fewer defecations per week were the most often used criteria for diagnosis of FC (34.8% (741/3628) and 34.2% (729/3828) respectively)”. - please explain these numbers to me. What is the number 3628 and 3828! I'm confused! These numbers need to be explained and the percentages carefully counted !;
6. In table 1 there are numbers "113/445, 677/1512, 91/237" and according to Figure 1 it should probably be "113/446, 677/1515, 91/238!";
7. In table 1 there are also numbers: 431, 1508, 192 and 2131 which I don't know where they come from or how they are counted !;
8. Please explain the number 1385 that appears on lines 168, 170, and 171;
Line 173-180 does not match numbers and percentages. What's the value of 2176 ?;
Line 177 is "Figure 3" should be "Figure 4";
Line 182 is "Figure 3" should be "Figure 4";
Line 183-196 in Table 2, please check the values as they do not agree with Figure 1 (1888 and 403?)!
Line 203 and 216 please check the numbers and percentages !;
Line 219 is "Figure 4" should be "Figure 5";
Line 227-228 is "Figure 4" should be "Figure 5";
9. No caption for Table 3;
10. The "Question" table does not contain the same questions as in “General Questions”? Is this right?;
Author Response
Brussels, 16 September 2022
To:
Prof. Dr. Maria Luz Fernandez
Editor in Chief
Section: Nutritional Epidemiology
Nutrients
Re: Manuscript titled “Reported prevalence and nutritional management of functional constipation among young children from healthcare professionals in eight countries across Asia, Europe and Latin America”
Dear Prof. Fernandez,
Thank you for sharing the further comments from the 2nd reviewer on the above-mention manuscript and for allowing us to submit a further revised version to Nutrients.
We would like to thank the reviewer for their remarks and constructive comments. The manuscript has been revised further to address the points raised by the reviewer. All changes are traceable in the word documents in highlighted statements. Please see our replies to all specific comments attached. We hope that the revised version of the manuscript is acceptable for publication in Nutrients soon.
We look forward to your favorable response.
Yours sincerely,
Prof Dr Yvan Vandenplas
Corresponding authors
Point-by-Point Response to Reviewers’ Comments
Reviewer: 1
|
Section |
Reviewer’s Comments |
Remarks |
|
Overview |
This review covers too few countries. Is there no problem in other countries, to which the authors of this article are devoted?
|
The review covers 8 countries from 3 different regions in the world with a total number of respondents of 2199. As such, we believe that the information presented in the manuscript will be beneficial for the rest of the countries in the world. |
|
Authors’ list |
Line 6 is "Marion M Aw 5,6 " should be "Marion M Aw5,6" (no spaces);
Lines 6-7 is "Badriul Hegar 8 " should be "Badriul Hegar8" (no spaces);
Line 10 and 22 are too large spaces;
Line 10 is: "the Ministry" should be: "the Ministry"
|
Revisions have been made |
|
|
Line 22 is “Gastrointestinal Motility. Gastroenterology ”should be“ Gastrointestinal Motility. Gastroenterology ";
|
Revision has been made |
|
Introduction |
Line 50 is "at least two of the following" should be "at least two of the following";
|
Revision has been made |
|
|
Line 51 must be deleted “;
Line 53 is "mass in the rectum 1 ." should be "mass in the rectum1.";
Line 55-56 is “obstruct the toilet 1 . "It should be" obstruct the toilet1. ";
Line 58 is "1.3% and 26.8% 2 ." it should be "1.3% and 26.8% 2.";
Line 59-60 is "(1-3 years of 59 age) 3-6 ." it should be “(1-3 years of 59 age)3-6.”;
Line 84 is' implementation. The name of the 'should be' implementation. The name of the ";
|
Revision has been made |
|
Methods |
Line 90-93 is “Each country was targeted to reach out to 100 respondents except for countries with limited number of HCPs such as Singapore and Hong Kong. Reminders were sent when the target number of respondents was not achieved." And Turkey also has less than 100 respondents (3% - 66!); |
Turkey has been added in the list of countries with less than 100 participants in the sentence in line 92. |
|
Results |
Line 107-111 is “The drop-out was due to exclusion criteria of having incomplete responses Respondents were from the following countries / regions: Russia (65.9%, 1449), Indonesia (11.0%, 242), Malaysia (6.0%, 132 ), Mexico (5.7%, 125), Kingdom of Saudi Arabia (KSA) (5.1%, 113), Turkey (66, 3.0%), Hong Kong (2.2%, 49), Singapore (1.0%, 23)” and in line 33-34: (Russia (67%!), Indonesia (11%), Malaysia (6%), KSA (5%!), Mexico (5.7%), Turkey (3%), Hong Kong (2.2%), Singapore (1%)). - It needs to be standardized !!!;
|
The information in now line 108-113 has been standardized and aligned with the information in now line 33-35 |
|
|
Line 117-111 is “The drop-out was due to exclusion criteria of having incomplete responses Respondents were from the following countries / regions: Russia (65.9%, 1449), Indonesia (11.0%, 242), Malaysia (6.0%, 132 ), Mexico (5.7%, 125), Kingdom of Saudi Arabia (KSA) (5.1%, 113), Turkey (66, 3.0%), Hong Kong (2.2%, 49), Singapore (1.0%, 23)". Please count the% again for KSA, Hong Kong and Singapore
I do not agree with the numbers and percentages in the whole work !!! You must check it all again! This misleads the person reading it.
|
All the numbers and percentages have been further checked for their accuracy. |
|
|
Figure 1 is hardly legible!
Line 117 is “Pediatricians represented the highest percentage of respondents (80.3%, 1759/2199)” and should be: “Pediatricians represented the highest percentage of respondents (80.3%, 1766/2199!)”;
|
Revisions to the figure and percentage have been made. |
|
|
In Figure 2, “Profession”, the percentages need to be corrected. For example, the text says 80.3% and the chart 80%; 6.3% and the figure 6.32% !; |
The percentage in the figure and text has been checked and aligned. The percentage of pediatricians has been revised to 80.3% in the figure and text. |
|
|
Line 128-129 is “Overall, the most often reported prevalence of toddler's FC was less than <5% (43.3%, 952/2189)” should be “Overall, the most often reported prevalence of toddler's FC was less than <5% (43.3%, 952/2199) ”;
|
Revision has been made as follows: “Overall, almost half of the respondents (43.3%, 952/2189). reported to estimate the prevalence of toddler’s FC was less than <5%.” |
|
|
Line 133 is "(Figure 2)" should be "(Figure 3)";
Line 137 is "(Figure 2)" should be "(Figure 3)";
Line 135 is "(1273/2199)" should be "((1275/2199)"; |
Revisions have been made |
|
|
Line 141 and 143 you have to count the percentages !;
|
The percentages have been re-checked and aligned |
|
|
Line 146 is "Malaysia, and Singapore" should be "Malaysia and Singapore";
|
Revisions have been made |
|
|
Line 146 is "44.8 % and 38.4% vs 25.4%" should be "44.8% and 38.4% vs 25.4%";
|
Revisions have been made by deleting the extra space |
|
|
Figure 3 is hardly legible. I don't understand this drawing. Please describe it more in the text;
Line 150-155 is “Forty-eight percent of the respondents (1014/3628) reported that all of the ROME IV criteria were important for the diagnosis of FC. Those who always used the ROME IV criteria were more likely to select all of the ROME IV criteria equally important than those who never used them (p<0.001). In addition, painful or hard bowel movements and two or fewer defecations per week were the most often used criteria for diagnosis of FC (34.8% (741/3628) and 34.2% (729/3828) respectively)”. - please explain these numbers to me. What is the number 3628 and 3828! I'm confused! These numbers need to be explained and the percentages carefully counted !;
|
Figure 3 has been revised to improved its readability. The denominator for each response can be varied due to the following: 1). Missing responses, 2) multiple answers. This information has been further clarified in the Methods section in now lines 97-99. |
|
|
In table 1 there are numbers "113/445, 677/1512, 91/237" and according to Figure 1 it should probably be "113/446, 677/1515, 91/238!"; |
These numbers are correct. Thus the total number 2199 is incorrect and has been revised to 2194. |
|
|
In table 1 there are also numbers: 431, 1508, 192 and 2131 which I don't know where they come from or how they are counted !; |
These denominator numbers are based on the number of total responses from the respondents. Thus, the total responses for the question- “Frequency of using ROME IV.. “” was 445, while the total responses for the question- “ The ROME IV criteria which is used most frequently were 431/1508/192. The total number of responses for this question was less than the first question. |
|
|
Please explain the number 1385 that appears on lines 168, 170, and 171; |
The way in which HCP could answer this question was different for Russia (one answer) and the other countries (multiple answers possible). Therefore, the denominator 1444 for Russia reflects the number of respondents, whereas the denominator 1385 for the other countries reflects all answers given. This has been further clarified and checked in the text. |
|
|
Line 173-180 does not match numbers and percentages. What's the value of 2176 ?;
|
The numbers have been further checked as the percentages were the round-up from two decimal points to one decimal point. The denominator number is based on the number of total responses from the respondents. |
|
|
Line 177 is "Figure 3" should be "Figure 4";
Line 182 is "Figure 3" should be "Figure 4";
|
Revisions have been made |
|
|
Line 183-196 in Table 2, please check the values as they do not agree with Figure 1 (1888 and 403?)!
|
The numbers have been further checked as the percentages were the round-up from two decimal points to one decimal point. The denominator number for Table 2 is different from Figure 1 due to the difference in the number of total responses for each question. The numbers have slightly changed, because missing values were taken out of the total number of responses. This is now consistent throughout the manuscript. |
|
|
Line 203 and 216 please check the numbers and percentages !;
|
The numbers have been further checked. |
|
|
Line 219 is "Figure 4" should be "Figure 5";
Line 227-228 is "Figure 4" should be "Figure 5";
|
Revisions have been made |
|
|
No caption for Table 3; |
Caption for Table 3 has been added as follows: “Overview of reported prevalence of FC based on epidemiological studies as compared to the current survey results” |
|
Appendix |
The "Question" table does not contain the same questions as in “General Questions”? Is this right?; |
Appendix 1 on the overview of the Questions by country has been revised to make it consistent with the questionnaire in Appendix 2.
|

Reviewer 2 Report
The authors displayed the prevalence and nutritional management of FC in toddlers. The study is interesting.
1. Could the authors specify why they choose these eight countries?
2. Could the authors explain the numbers and values in Line 128. "Overall, the most often reported....", in this sentence, what is 5% and what is 43%, it is not consistent with the figure 2. I think it is confusing.
3. In figure 1, authors showed P<0.001 between regions. it is totally unclear. How the authors compare since there are different ranges in different regions. Tha authors need to specify it.
4. Line 119, 43% respondents have 15 years' experience. but the authors stated, "most respondents", is 43% is most?
5. there are some mistakes in the whole text such as line 108, please check it carefully.
Author Response
|
Section |
Reviewer’s Comments |
Remarks |
|
Overview |
The authors displayed the prevalence and nutritional management of FC in toddlers. The study is interesting.
|
Thank you for the kind remarks |
|
Methods |
Could the authors specify why they choose these eight countries? |
The information is mentioned in the Methods section on lines 71-72.
|
|
|
Could the authors explain the numbers and values in Line 128. "Overall, the most often reported....", in this sentence, what is 5% and what is 43%, it is not consistent with figure 2. I think it is confusing. |
The statement has been revised for better clarity as follows: “Overall, almost half of the respondents (43.3%, 952/2189). reported to estimate the prevalence of toddler’s FC was less than <5%.” |
|
|
In figure 1, authors showed P<0.001 between regions. it is totally unclear. How the authors compare since there are different ranges in different regions. The authors need to specify it. |
The p value has been removed to avoid confusion. |
|
|
Line 119, 43% respondents have 15 years' experience. but the authors stated, "most respondents", is 43% is most? |
The statement has been revised as follows: “Almost half of the respondents (43.0%, 946/2195) had more than 15 years of experience, while 24.0 % (528/2159) had less than 5 years experience and the rest were in-between these periods (Figure 2). |
|
|
there are some mistakes in the whole text such as line 108, please check it carefully. |
A full stop has been added to improve the clarity of the statement. |

Round 2
Reviewer 1 Report
I followed the corrections to my comments. They have been plotted. The reader may, however, make a mistake in some places with the percentages, because you have to read carefully and count. As for the respondents, I am still in favor of the fact that few countries are included in this review. The majority, as many as 67%, are respondents from Russia.

Author Response
Response to reviewer
Thank you for the further comments. We have added the information on the majority of respondents are from Russia in the abstract (line 34) and results section (line 114).
Reviewer 2 Report
No further suggestions.
Author Response
Thank you for the kind confirmation.